# Impact of a Clinical Decision Support System on the Change over Time in the Anticholinergic Load in Geriatric Patients: The SADP-Antichol Study

**DOI:** 10.3390/pharmacy12060162

**Published:** 2024-10-30

**Authors:** Grégoire Delvallée, Lisa Mondet, Chloé Cornille, Guillaume Deschasse, Aurélie Lenglet

**Affiliations:** 1Pharmacy Service, Centre Hospitalo-Universitaire Amiens-Picardie, F-80000 Amiens, France; delvallee.gregoire@chu-amiens.fr (G.D.); mondet.lisa@chu-amiens.fr (L.M.); cornille.chloe@chu-amiens.fr (C.C.); 2Geriatric Service, Centre Hospitalo-Universitaire Amiens-Picardie, F-80000 Amiens, France; deschasse.guillaume@chu-amiens.fr; 3Mécanismes Physiopathologiques et Conséquences des Calcifications Cardiovasculaires (MP3CV) Laboratory, Centre Universitaire de Recherche en Santé, Centre Hospitalo-Universitaire Amiens-Picardie, F-80000 Amiens, France

**Keywords:** clinical pharmacy, cholinergic antagonists, clinical decision support system, deprescribing

## Abstract

Purpose: Anticholinergic drugs can cause adverse events (AEs) in older adults. Clinical decision support systems (CDSSs) can detect prescriptions with a high anticholinergic load. Our starting hypothesis was that the anticholinergic load could be reduced by combining a CDSS with a strategy for generating pharmacist interventions. The objective of the present study was to assess the impact of this combination on the change over time in the anticholinergic load in hospitalized older adults. Methods: This prospective, single-centre study was divided into two 6-week periods. During the interventional period, a pharmacist analyzed the alerts generated by the CDSS for 30 targeted anticholinergic drugs and decided whether to issue a pharmacist intervention. A control period corresponds to standard care. The primary endpoint of the study is the delta of the anticholinergic load between the alert and hospital discharge; the secondary endpoint is the incidence of anticholinergic adverse events (AEs). Results: Of the 144 alerts generated, 87 were considered to be relevant (36 in the interventional period and 51 in the control period). A significant difference was observed between the delta anticholinergic load between the experimental and control periods (1.61 vs. 0.67, *p*-value = 0.0115). For the targeted drugs (*n* = 94) over the 87 alerts, 46.8% were for antihistamines and 21.3% were for desloratadine. Of the 36 pharmacist interventions sent by the pharmacist, 19 (52.8%) were accepted. The most deprescribed drug class was the antihistamine class (*n* = 7), and the most deprescribed drug was amitriptyline (*n* = 5). Among these 87 patients with alerts, the correlation between the anticholinergic load and the number of AEs was not statistically significant (*p* = 0.887). The most common AE affecting the peripheral nervous system was constipation (28.6%), and the most common AE affecting the central nervous system was confusion (29.9%). Conclusions: Our results showed that the combination of specific CDSS rules with pharmacist-mediated risk management procedures could further reduce the anticholinergic load in hospitalized older adults, relative to routine care. It remains to be determined whether this reduction in the anticholinergic load has an impact on the incidence of peripheral and central anticholinergic AEs, and thus the health of these patients.

## 1. Introduction

Anticholinergic drugs can cause adverse events (AEs) in older adults. These drugs are widely prescribed for the management of various conditions, including depression, epilepsy seizures, neuropathic pain, asthma, overactive bladder and ophthalmic disorders [1]. Anticholinergic syndrome (also referred to as atropine syndrome) corresponds to the set of effects produced when acetylcholine receptors are blocked by an antagonist such as atropine. When several anticholinergic drugs are combined, their cumulative effects are referred to as the anticholinergic load. Although some of atropine’s effects are therapeutic, others (especially when unexpected) are dangerous for the patient’s health and must be avoided. The risk of cognitive impairment is twice as high in older adults who take anticholinergic drugs on a regular basis than in those who do not [2].

Anticholinergic drugs cause tachycardia, urine retention, constipation, xerostomia, xerophthalmia, bronchodilatation and mydriasis by affecting the peripheral nervous system. In the central nervous system, confusion, disorientation, delirium, hallucinations, agitation, irritability, aggressiveness, amnesic disorders and dementia may be signs of anticholinergic syndrome [3].

The anticholinergic load can be quantified on scales that classify drugs according to their anticholinergic potential [4,5,6]. These scales can be used to measure the cumulative anticholinergic load imposed by drugs on the cholinergic system, by adding up the score for each drug (which ranges from 0 to 3, or sometimes 4) [5,6,7].

Polypharmacy (defined as the regular use of five or more medications at the same time) is associated with an elevated iatrogenic risk, a greater likelihood of hospital admission and higher mortality rates among older adults [8,9]. Furthermore, polypharmacy accentuates the risks of drug–drug interactions, drug–supplement/nutraceutical interactions and drug–illness interactions [10].

Computerized clinical decision support systems (CDSS) have been developed to automatically detect high-risk drug situations and prioritize pharmacist interventions (PIs). The CDSSs provide automated near-real-time alerts on potentially inappropriate prescriptions to older adults, based on the data (on the drug prescribed, the patient’s age and blood test results) from the computerized physician order entry (CPOE) system. Today’s methods of reducing the iatrogenic risk (e.g., the expert review of prescriptions) are time-consuming and costly in terms of human resources. Despite the recent, massive computerization of the prescribing process, CDSSs have yet to be optimized in practical terms. If a CDSS is to be used on a routine basis, alerts about the anticholinergic load must be developed. This construction process involves the transformation of implicit knowledge (clinical experience, published scores, etc.) into explicit rules that combine the characteristics (age, the medications taken, laboratory test results, etc.) available in medical informatics systems. Ideally, a CDSS should be able to alert healthcare professionals to at-risk situations in real time. Our starting hypothesis was that the combination of a specific CDSS rule with a strategy for managing and transmitting alerts would reduce the anticholinergic load of prescriptions for hospitalized older adults and thus prevent the occurrence of adverse effects linked to anticholinergic syndrome.

The study’s primary objective was to determine whether the combination of a computerized clinical decision support system with an alert management procedure was associated with a reduction in the anticholinergic load in hospitalized older adults, relative to a standard procedure.

The secondary objective was to identify the occurrence of adverse events and determine whether the latter was correlated with the anticholinergic load.

## 2. Methods

This prospective, single-centre study was conducted at Amiens University Hospital (Amiens, France). In line with the French legislation on studies of routine clinical practice, the study protocol was approved by a hospital committee with competency for research not requiring authorization by an institutional review board (reference: PI2023_843_0010) and complied with the French government’s MR-004 reference methodology governing the processing of personal data for research purposes.

The study was carried out in three phases: (i) technical implementation of the rules in the CDSS and creation of alert management procedures; (ii) a “test” phase to check that the data circuit was compliant; and (iii) an intervention phase during which the two groups were compared.

The first phase focused on the implementation of procedures for creating alerts that reliably identified hospitalized older adults whose anticholinergic load on admission is non-zero and potentially reducible (the criteria are detailed below). An alert may concern several anticholinergic drugs. The procedure must generate an alert and an associated PI. A course of action will be suggested to the pharmacist, who reviews the CDSS alert and checks that it is relevant to the patient’s specific clinical situation. A standardized PI may be generated and sent to the prescriber. The CDSS is a decision support tool and can be adapted to suit specific clinical situations. The rule implemented in the CDSS focused on drugs with anticholinergic activity listed on the French Coefficient d’Imprégnation Anticholinergique (CIA) scale [11] and the Spanish Cribado de Deterioro Cognitivo (CRIDECO) [12] scale: the 30 targeted drugs are listed in the Appendix A. We selected drugs with different scores (1, 2 and 3) from drug classes that enabled therapeutic reassessment without necessarily requiring specialist advice (antihistamines, for example) or for which therapeutic reassessment is carried out in routine practice (psychotropic drugs, for example). In contrast, we did not include drugs in therapeutic classes that require specialist advice or therapeutic titration (immunosuppressants and anti-epileptics, for example). All the drugs used to set the alert parameters have a high anticholinergic load and can be re-evaluated during the hospital admission. Deprescribing was recommended for all drug classes, although alternatives were sometimes suggested (urinary antispasmodics and antihistamines).

The initial cohort consisted of 144 patients. The study’s main inclusion criteria were an age of 75 or over and the admission to a medical, surgical or obstetric department in the hospital. The age threshold (75 or over) was defined with regard to the French healthcare system and the literature data [13]. The exclusion criteria were death, admission to a palliative care unit or day hospital, or a length of hospital stay below 3 days.

During the second (test) phase, the case files of 10 patients were screened to check the validity of the inclusion on CDSS alerts (patients aged 75 or over, hospitalized in a care unit and whose initial prescription included at least one of the targeted molecules) and ensure compliance with the data circuit. A PI was generated but not sent to the prescriber. The data collected in the test phase were not analyzed.

The third phase comprised two distinct periods (Figure 1) of 6 weeks each: an interventional period during which the CDSS was combined with a strategy for managing and transmitting alerts concerning the anticholinergic load, and a control period of routine care. During both periods, the patient was seen on the ward by a physician, and prescriptions were issued through the CPOE system.

During the interventional period, all prescriptions were analyzed by the CDSS Pharmaclass^®^ (version 2.1.7.1, Keenturtle, Lyon, France), which could generate (or not, depending on the rules) an alert and sent it to the pharmacist. The latter analyzed the relevance of the alert and decided to trigger (or not) a PI, according to a standardized procedure. This PI was communicated to the prescriber verbally and/or through the CPOE system. The physician then accepted or refused the PI. The CDSS analyzed the prescriptions in real time and was updated whenever a computerized prescription was changed.

During the control period (routine care), a prescription was analyzed by a pharmacist if the patient was admitted to a department where prescriptions are analyzed (40% of the hospital’s prescriptions) and may lead to a PI transmitted to the prescriber orally and/or by computer. This could be performed at any time. During this phase, the computerized prescription is analyzed by the CDSS, but the alert is not transmitted to the pharmacist.

During both periods, the clinical pharmacist discussed changes in treatment with prescribers; this was carried out verbally or through the CPOE. The clinical pharmacist issuing PIs practised in a geriatric care unit. In collaboration with other healthcare staff, clinical pharmacists provide drug information services to a wide variety of prescribers; their specialist services may include therapeutic drug monitoring, involvement in clinical drug toxicology, and pharmacovigilance.

The study’s primary endpoint was the change over time in the patient’s total anticholinergic load. This was calculated once when the alert was generated in the CDSS and again when the patient was discharged, on the basis of the discharge prescription. The anticholinergic load per drug ranged from 0 to 3.

To address the study’s primary objective, the interventional group and the control group were compared with regard to the mean change over time in anticholinergic load. Since the data for these variables were not normally distributed, a Kruskal–Wallis nonparametric test was applied. The normality of the distribution of quantitative variables was checked with a Shapiro–Wilk test. Intergroup comparisons were performed with a chi-square test (for categorical variables), and Student’s *t* test or Mann–Whitney test (for continuous variables). The threshold for statistical significance was set to *p* < 0.05.

To address the study’s secondary objectives, information on anticholinergic AEs was extracted from each patient’s medical record: emergency department or hospital admission reports, consultation notes, imaging and laboratory reports, and vital signs. These AEs were classified as affecting the peripheral nervous system (tachycardia, acute urine retention, constipation, xerostomia, xerophthalmia, and mydriasis) or the central nervous system (confusion, disorientation, agitation, irritability, aggressiveness, and memory disorders). A Pearson’s correlation test was used to assess the correlation between the anticholinergic load and the number of AEs experienced by the patients in the study. Again, the threshold for statistical significance was set to *p* < 0.05.

## 3. Results

### 3.1. Characteristics of the Study Population

The CDSS generated a total of 144 alerts during the study period as a whole, corresponding to 144 patients: 56 (mean age: 83.1 ± 6.6; women: 57.1%) were selected during the interventional period, and 88 (mean age: 82.5 ± 5.8; women: 55.7%) were selected during the control period. Of the 144 patients, 105 (72.9%) were in medical wards and 39 (27.1%) were in surgical wards. The mean ± standard deviation (SD) of initial anticholinergic load (calculated at the time when the CDSS generated the alert) was 3.94 ± 1.48. On average, each patient taking at least one compound with anticholinergic activity was receiving 1.58 drugs with a score of 1, 1.02 drugs with a score of 2, and 1.06 drugs with a score of 3. Of the drugs targeted by the alerts, 75.7% had been prescribed prior to the patient’s admission to hospital (Table 1).

Analysis of the patients’ medical records and verification of the eligibility criteria resulted in the selection of 36 alerts during the interventional period and 51 alerts during the control period. The corresponding anticholinergic AEs were also analyzed (Figure 2).

Most of the alerts that were not selected were linked to a clinical context that prevented the reassessment of the targeted drug (short-term prescriptions, and long-term prescriptions with follow-ups by a specialist). The alerts that were not selected concerned cetirizine (52%), hydroxyzine (14%), and amitriptyline (14%) in particular. The other clinical contexts that rendered alerts irrelevant were palliative/end-of-life care, short hospital stays that prevented reassessment of the patient’s medication, drugs prescribed on an “if necessary” basis but not administered, and drugs discontinued following a medication review.

### 3.2. Changes over Time in the Anticholinergic Load

During the interventional period (*n* = 36), the mean anticholinergic load was 4.47 ± 1.42 at the time of the alert and 2.86 ± 2.13 at discharge; this corresponded to a mean reduction of 1.61 ± 1.92 (Figure 3). During the control period (*n* = 51), the mean anticholinergic load was 4.24 ± 1.53 at the time of the alert and 3.57 ± 1.84 at discharge; this corresponded to a mean reduction of 0.67 ± 1.49 (Figure 4). The difference in the change over time in the anticholinergic load between the intervention group and the control group was statistically significant (*p*-value = 0.015).

For alerts for which the PI was accepted (*n* = 19), the mean ± SD reduction in the anticholinergic load was 3.16 ± 1.30.

### 3.3. Pharmacist Interventions

The 87 relevant PIs concerned 94 drug prescriptions; these were mainly antihistamines (46.8%) (Figure 5). Desloratadine (21.3%) was the most frequently involved individual drug (Figure 6). Other involved drugs were trospium (2.1%), clomipramine (2.1%), oxybutynin (1.1%), cimetidine (1.1%), fexofenadine (1.1%), trihexyphenidyl (1.1%) and loratadine (1.1%).

Of the 36 relevant PIs issued during the interventional period (*n* = 36), 19 (52.8%) were accepted by the prescriber. The most deprescribed drug class was antihistamines (*n* = 7) (Figure 7) and the most deprescribed individual drug was amitriptyline (*n* = 5) (Figure 8). The reasons for the non-acceptance of PIs were not known, because the physicians were not obliged to explain them.

### 3.4. Anticholinergic Adverse Events

When considering the 87 patients for whom a relevant PI was issued, the mean number of anticholinergic AEs affecting the peripheral or central nervous system was 2.1 ± 1.8 per patient in the interventional group and 1.6 ± 1.9 per patient in the control group (Figure 9). The most common peripheral AE was constipation (28.7%) (Figure 10), and the most common central AE was confusion (29.9%) (Figure 11). According to Pearson’s correlation test, the anticholinergic load at the time of the alert was not significantly associated with the number of AEs (*p*-value = 0.887).

## 4. Discussion

The results of our study of hospitalized older adults highlighted a reduction in the anticholinergic load between the time of the CDSS alert and the discharge prescription. The mean reduction in anticholinergic load was significantly greater during the interventional (CDSS) period than during the control (standard care) period (1.61 vs. 0.67, respectively). The CDSS was useful for detecting prescription of anticholinergic drugs for older adult patients. The harmonized procedures for each drug class made it possible to structure the PI and thus prompt the deprescription or replacement of drugs with a high anticholinergic load.

In our study, 60.4% of the alerts led to a PI. Several literature reviews have suggested that the routine use of a CDSS has only a small clinical impact on older adult patients and, indeed, patients in general [14]. However, the alerts generated in the literature were often simple and “static”, without an expert filter or human intervention to determine the relevance of the alert. Hence, a high proportion of the alerts might have been of little relevance to the patient’s specific clinical situation. In the majority of studies, alerts are transmitted directly to the prescriber, and an alternative treatment is not suggested [15,16,17,18]. Although it has been shown that the involvement of clinical pharmacists improves the acceptance of CDSS alerts, pharmacists are rarely involved in alert management. A PI alerting the prescriber to the total anticholinergic load on admission might lead to a greater reduction in the load than a conventional medication review [19]. Prescribers involved in medication reviews must also be advised not only on pharmacodynamic drug interactions but also on pharmacokinetic interactions described in the literature [20,21,22].

Despite the encouraging result on variation, the PI acceptance rate in our study was only 52.8%; this value is at the low end of the range described by a literature review (52% to 83%; the highest values were recorded when the pharmacist was present on the ward) [23]. This low value can be explained by the fact that prescribers were sometimes unaware that PIs were also sent via the CPOE system; verbal transmission was recommended, but was unsuccessful when the prescriber was not available. It might be possible to increase the PI acceptance rate by collaboration with the clinical pharmacologist. In fact, clinical pharmacologists have an important role in improving the safe, effective use of established drugs [24]. The accuracy of the anticholinergic burden rating could be increased by considering pharmacokinetic and pharmacodynamic factors such biophase drug concentrations, the pharmacologically active metabolites formed after drug administration, and muscarinic-receptor-mediated effects. The reasons for vulnerability to anticholinergic adverse events in older adults include significant functional changes in organ systems with ageing, chronic and comorbid conditions, and polypharmacy [25].

We observed that 28.7% of the study participants suffered from constipation and 29.9% suffered from confusion. Several studies have highlighted the limited effectiveness of PIs in reducing the anticholinergic load [26,27] and the incidence of adverse drug reactions in older adults [28,29]. In the DEFEAT-polypharmacy randomized clinical trial, 72% of the PIs linked with the deprescription of anticholinergic drugs were accepted by general practitioners. This led to a reduction in falls, but not an improvement in cognition or greater self-reported quality of life.

A recent literature review [30] found that although only two of the randomized studies of deprescribing anticholinergic drugs had been carried out in hospital settings, both found a reduction in anticholinergic AEs.

The medications monitored in our study were those listed on the French CIA scale [11]), whereas many of the other scales described in the literature [31] feature the medications prescribed in the USA. The CIA scale was developed by considering both in vitro data on serum anticholinergic activity and expert data from clinicians. It includes the drug classes most commonly found in the literature: psychotropic drugs, antiparkinsonian agents, antihistamines, analgesics, urinary antispasmodics, etc. A drug not featured in the CIA scale (fesoterodine) was added to our list because it is frequently prescribed in France, as indicated in the literature from 2015 [7] and 2018 [32].

Our pilot study had several limitations. Firstly, it had limited statistical power. Secondly, it was a single-centre study. Although we observed a significant reduction in the anticholinergic load, our hypothesis should now be confirmed by extending the use of the CDSS to other establishments (in both in-hospital and out-patient settings) and include older adult patients with polypharmacy. Thirdly, we studied only one anticholinergic burden scale, even though many such scales have been described in the literature [4,7]. However, we studied a scale that was best suited for use in a French hospital environment.

The anticholinergic AEs in our study population were described retrospectively by analyzing the patients’ electronic medical records. This process depended on the information provided by the medical staff, and so the incidence of these AEs was probably underestimated. For example, xerophthalmia can be detected by prescribing an ocular lubricant—a drug class that is often not included in home treatments.

To improve the impact of PIs, the transmission method needs to be formalized in the CPOE and by verbal transmission; prescribers need to be made aware of the clinical pharmacist’s opinion.

## 5. Conclusions

Our results showed that the combination of specific CDSS rules with pharmacist-mediated risk management procedures can further reduce the anticholinergic load in hospitalized older adults, relative to routine care. It remains to be determined whether this reduction in the anticholinergic load has an impact on the incidence of peripheral and central anticholinergic AEs, and thus on the health of these patients.

## Figures and Tables

**Figure 1 pharmacy-12-00162-f001:**
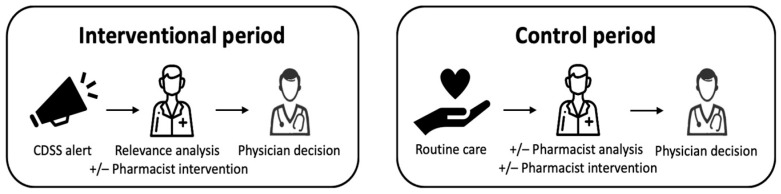
The working processes for the two distinct study periods.

**Figure 2 pharmacy-12-00162-f002:**
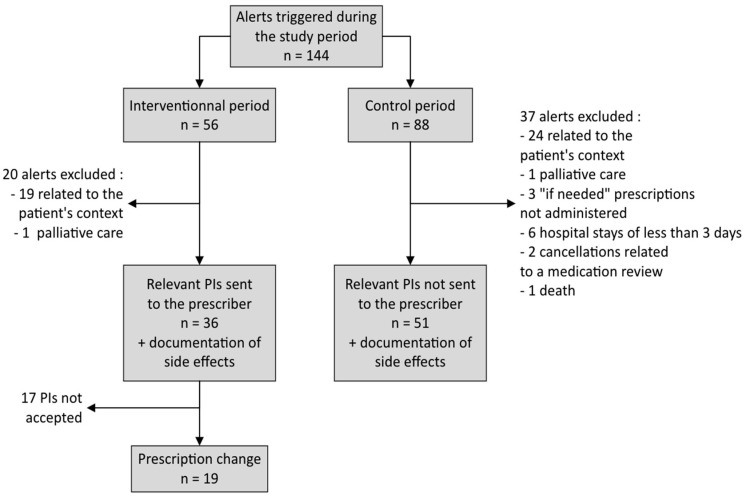
Study flowchart.

**Figure 3 pharmacy-12-00162-f003:**
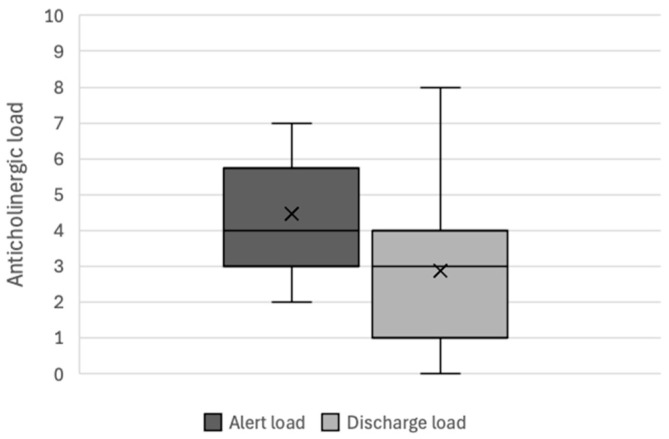
Changes in the total anticholinergic load during the interventional period (*n* = 36).

**Figure 4 pharmacy-12-00162-f004:**
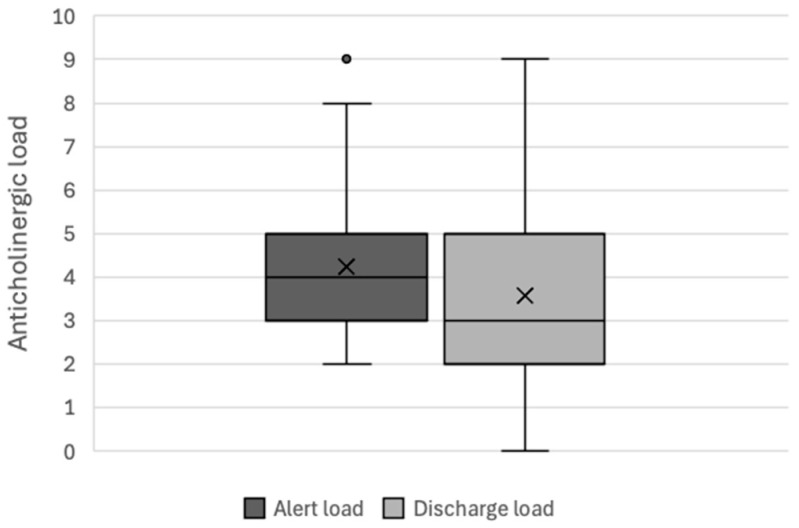
Changes in the total anticholinergic load during the control period (*n* = 51).

**Figure 5 pharmacy-12-00162-f005:**
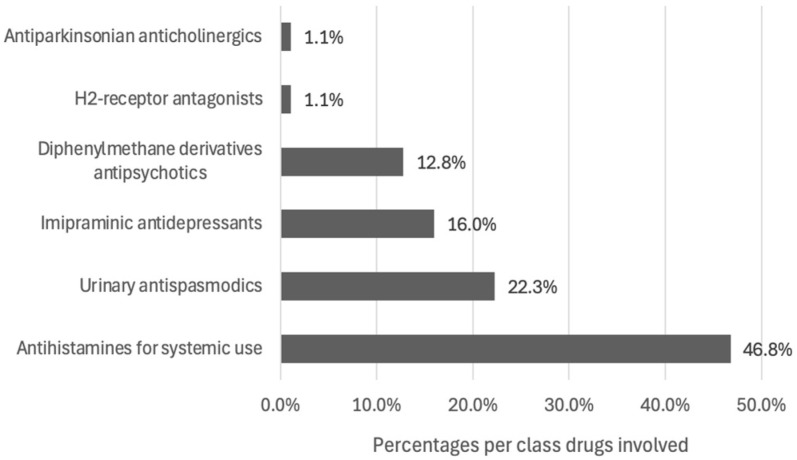
The drug classes targeted by PIs (*n* = 94).

**Figure 6 pharmacy-12-00162-f006:**
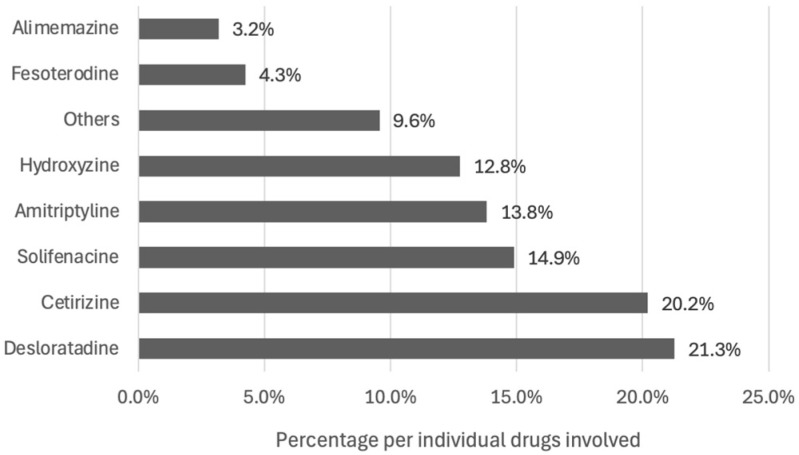
The individual drugs targeted by PIs (*n* = 94).

**Figure 7 pharmacy-12-00162-f007:**
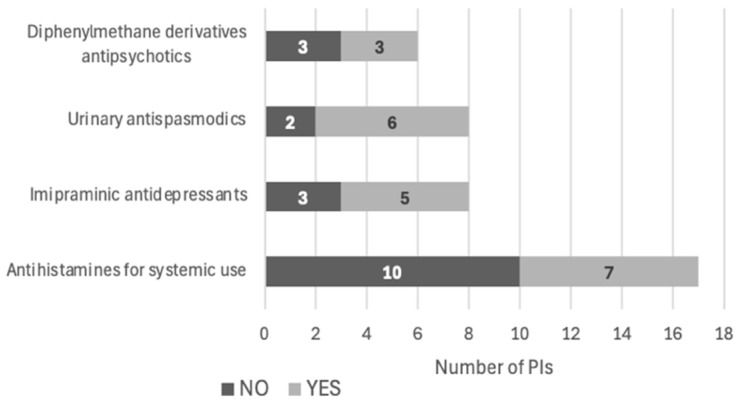
PI acceptance by drug class (*n* = 39).

**Figure 8 pharmacy-12-00162-f008:**
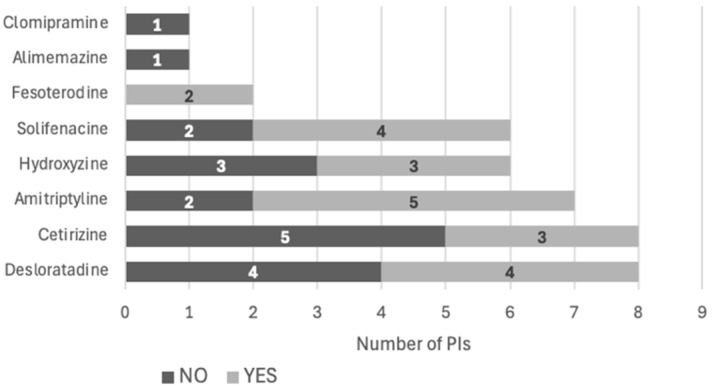
PI acceptance by individual drug (*n* = 39).

**Figure 9 pharmacy-12-00162-f009:**
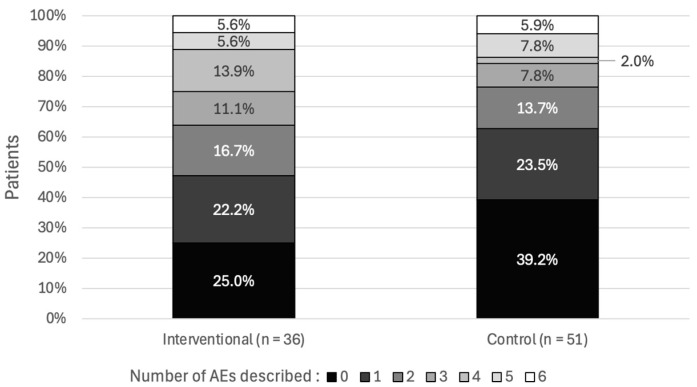
The number of anticholinergic AEs in patients with alerts (*n* = 87).

**Figure 10 pharmacy-12-00162-f010:**
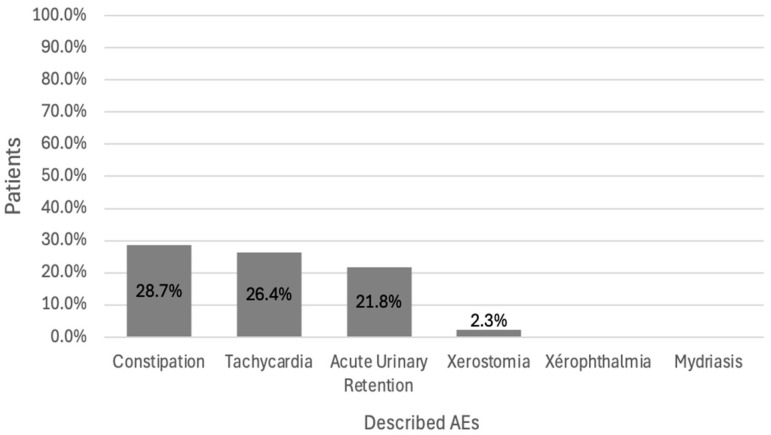
Types of peripheral anticholinergic AEs in patients with alerts (*n* = 87).

**Figure 11 pharmacy-12-00162-f011:**
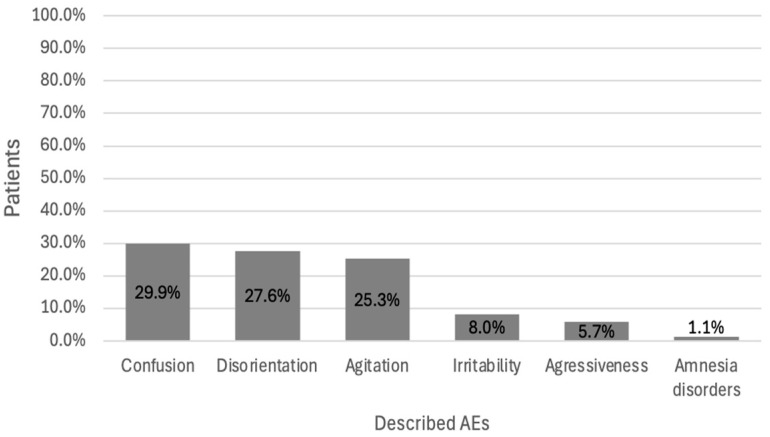
Types of central anticholinergic AEs in patients with alerts (*n* = 87).

**Table 1 pharmacy-12-00162-t001:** Characteristics of the study population.

	Interventional (*n* = 56; 38.9%)	Control (*n* = 88; 61.1%)	Total (*n* = 144; 100%)	*p*
Age (years)	83.1 ± 6.6	82.5 ± 5.8	82.7 ± 6.1	0.069
Female sex	32 (57.1%)	49 (55.7%)	81 (56.4%)	0.863
Type of hospital ward				
Medical	42 (75.0%)	63 (71.6%)	105 (72.9%)	0.654
Surgical	14 (25.0%)	25 (28.4%)	39 (27.1%)	
Anticholinergic drugs				
Anticholinergic load at the time of the alert	4.05 ± 1.52	3.86 ± 1.46	3.94 ± 1.48	0.552
Number of drugs per patient:				
with a score of 1	1.74 ± 1.01	1.47 ± 1.01	1.58 ± 0.93	0.031
with a score of 2	1.00 ± 0.50	1.03 ± 0.52	1.02 ± 0.51	0.487
with a score of 3	1.06 ± 1.55	1.05 ± 0.54	1.06 ± 0.54	0.975
Source of the prescription targeted by the alert				
Hospital prescription	11 (19.6%)	24 (27.3%)	35 (24.3%)	0.298
Home prescription	45 (80.4%)	64 (72.7%)	109 (75.7%)	

The data are quoted as the mean ± SD or n (%).

## Data Availability

The data presented in this study are available upon request from the corresponding author. The data are not publicly available due to academic privacy.

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
