# Peer review of "Impact of a Clinical Decision Support System on the Change over Time in the Anticholinergic Load in Geriatric Patients: The SADP-Antichol Study"

_pharmacy, 2024, doi:10.3390/pharmacy12060162_

Round 1
Reviewer 1 Report
Comments and Suggestions for Authors
The paper by Delvallée and colleagues is interesting and of strong actuality. Nevertheless, I have some major comments.
Introduction
The definition of polypharmacy intended as the concomitant consumption of 5 or more drugs must be included. Authors should discuss briefly the possibility of drug-drug interactions, drug-supplement/nutraceutical interaction and drug illness interaction.
Line 48: “drugs have the opposite effect to acetylcholine” it is not necessary, remove it.
Objectives
This section is not necessary. Please, add the text to introduction.
Methods
The authors must insert the number of total population from which patients were enrolled.
There are many different scales to evaluate the anticholinergic burden. The authors should include them in the analysis or must add a statement in the limitations section.
Lozano-Ortega G, Johnston KM, Cheung A, Wagg A, Campbell NL, Dmochowski RR, Ng DB. A review of published anticholinergic scales and measures and their applicability in database analyses. Arch Gerontol Geriatr. 2020 Mar-Apr;87:103885. doi: 10.1016/j.archger.2019.05.010.
Ramos H, Moreno L, Pérez-Tur J, Cháfer-Pericás C, García-Lluch G, Pardo J. CRIDECO Anticholinergic Load Scale: An Updated Anticholinergic Burden Scale. Comparison with the ACB Scale in Spanish Individuals with Subjective Memory Complaints. J Pers Med. 2022 Feb 3;12(2):207. doi: 10.3390/jpm12020207.
Lines 102-105: this section must be modified considering only inclusion and exclusion criteria. Remove the denomination “non-inclusion”.
Geriatric age is generally estimated to start around 65 years. Nevertheless, this cut-off is highly debated, considering the increase of lifetime duration. The authors should clarify and discuss their cut-off choice (75 years) in inclusion criteria.
Ouchi Y, Rakugi H, Arai H, Akishita M, Ito H, Toba K, Kai I; Joint Committee of Japan Gerontological Society (JGLS) and Japan Geriatrics Society (JGS) on the definition and classification of the elderly. Redefining the elderly as aged 75 years and older: Proposal from the Joint Committee of Japan Gerontological Society and the Japan Geriatrics Society. Geriatr Gerontol Int. 2017 Jul;17(7):1045-1047.
Line 133-134: the authors must write a statement about normality to justify the use of tests like Student’s t-test and explain which test they used to assess normality.
Furthermore, the authors must describe how the pharmacists relate with physician in modifying patients’ therapy and discuss (in the following sections) the necessity of a clinical pharmacologist (MD) in the team.
Results
Line 165: the use of ellipsis is not necessary, correct throughout the text.
Table 5: the authors must clarify in the text what “others” stands for.
Figure 9-10: remove the “non described” graphic representation.
Discussion
Discuss the possible role of the presence of MD Clinical Pharmacologist in the team alongside pharmacists and the necessity of a multidisciplinary board.
In general, the number of references is too low. The authors must increase it, especially in introduction and discussion. They have to increase the description of the different anticholinergic drug types and their clinical usage (introduction) and discuss more deeply the possible DDI related to anticholinergic drugs, including not only pharmacodynamic interactions, but also pharmacokinetic interactions (example: antihistamines and CYP2D6).
Author Response
Comment 1: Introduction
The definition of polypharmacy intended as the concomitant consumption of 5 or more drugs must be included. Authors should discuss briefly the possibility of drug-drug interactions, drug-supplement/nutraceutical interaction and drug illness interaction.
Response 1: The definition of polypharmacy and a paragraph on drug-drug interactions, drug-supplement/nutraceutical interaction and drug illness interactions (line 57) have been added, together with the following reference: Vrolijk MF, Opperhuizen A, Jansen EHJM, Bast A, Haenen GRMM. Anticholinergic Accumulation: A Slumbering Interaction between Drugs and Food Supplements. Basic Clin Pharmacol Toxicol 2015;117:427–32. https://doi.org/10.1111/bcpt.12437.
Furthermore, we have added (line 303) the following sentence to the Discussion: “Thirdly, we studied only one anticholinergic burden scale, even though many such scales have been described in the literature [4,7]. However, we studied a scale that was best suit-ed to use in a French hospital environment.”.
Comment 2: Line 48: “drugs have the opposite effect to acetylcholine” it is not necessary, remove it.
Response 2: This sentence has been deleted.
Comment 3: Objectives
This section is not necessary. Please, add the text to introduction.
Response 3: This section has been deleted and the text has been added to the Introduction.
Comment 4: Methods
The authors must insert the number of total population from which patients were enrolled.
Response 4: The number of patients in the initial cohort (n=144) has been added to this section (line 116)
Comment 5: There are many different scales to evaluate the anticholinergic burden. The authors should include them in the analysis or must add a statement in the limitations section.
Response 5: Thank you for your comment. We now cite additional reviews on anticholinergic scales from the recent literature (line 54). Our study was based on a single scale that we considered to be the most relevant for use in a French hospital environment. We now mention the use of a single scale as a study limitation (lines 303-305).
Comment 6: Lines 102-105: this section must be modified considering only inclusion and exclusion criteria. Remove the denomination “non-inclusion”.
Response 6: Admission to a palliative care unit or day hospital has been added as an exclusion criteria (line 118).
Comment 7: Geriatric age is generally estimated to start around 65 years. Nevertheless, this cut-off is highly debated, considering the increase of lifetime duration. The authors should clarify and discuss their cut-off choice (75 years) in inclusion criteria.
Ouchi Y, Rakugi H, Arai H, Akishita M, Ito H, Toba K, Kai I; Joint Committee of Japan Gerontological Society (JGLS) and Japan Geriatrics Society (JGS) on the definition and classification of the elderly. Redefining the elderly as aged 75 years and older: Proposal from the Joint Committee of Japan Gerontological Society and the Japan Geriatrics Society. Geriatr Gerontol Int. 2017 Jul;17(7):1045-1047.
Response 7: Thank you for pointing this out. A statement and the above reference have been added to this section (line 118).
Comment 8: Line 133-134: the authors must write a statement about normality to justify the use of tests like Student’s t-test and explain which test they used to assess normality.
Response 8: Thank you very much for this comment. We have added further details to the sections on statistics. The normality of the distribution of quantitative variables was checked with a Shapiro-Wilk test. Intergroup comparisons were performed with a chi-square test (for categorical variables), Student’s t test or Mann-Whitney test (for continuous variables) (line 157).
Comment 9: Furthermore, the authors must describe how the pharmacists relate with physician in modifying patients’ therapy and discuss (in the following sections) the necessity of a clinical pharmacologist (MD) in the team.
Response 9: We agree with you and now specify how pharmacists discuss changes in treatment with physicians (line 145).
Comment 10: Results
Line 165: the use of ellipsis is not necessary, correct throughout the text.
Response 10: We removed the ellipses from the text.
Comment 11: Table 5: the authors must clarify in the text what “others” stands for.
Response 11: « Others » has been detailed for seven drugs concerned. The respective percentages have been added (line 215).
Comment 12: Figure 9-10: remove the “non described” graphic representation.
Response 12: We have removed the legend from the graphic.
Comment 13: Discussion
Discuss the possible role of the presence of MD Clinical Pharmacologist in the team alongside pharmacists and the necessity of a multidisciplinary board.
Response 13: Thank you for this highly relevant suggestion. This subject is now included in the Discussion (line 272) and a sentence has been added:
“It might be possible to increase the PI acceptance rate by collaboration with the clinical pharmacologist. In fact, clinical pharmacologists have an important role in improving the safe, effective use of established drugs [22]”.
Comment 14: In general, the number of references is too low. The authors must increase it, especially in introduction and discussion. They have to increase the description of the different anticholinergic drug types and their clinical usage (introduction) and discuss more deeply the possible DDI related to anticholinergic drugs, including not only pharmacodynamic interactions, but also pharmacokinetic interactions (example: antihistamines and CYP2D6).
Response 14: We agree with these remarks. Hence, we have added text to the Introduction and briefly mention the literature on anticholinergic drugs. We have added several references, including the following literature review:
Ruxton K, Woodman RJ, Magoni AA. Drugs with anticholinergic effects and cognitive impairment, falls and all-cause mortality in older adults. A systematic review and meta-analysis. Br J Clin Pharmacol 2015; 80: 209–220.
We also discuss the pharmacokinetics and pharmacodynamics of anticholinergic drugs in more detail and have added several corresponding references.
Reviewer 2 Report
Comments and Suggestions for Authors
It is important to investigate the anticholinergic load could be reduced by combining a CDSS with a strategy for generating pharmacist interventions. However, I think that several issues need to be resolved.
1) In this study, a pharmacist analyzed the alerts generated by the CDSS for 30 targeted anticholinergic drugs (P1, LL16-17). However, the Methods section lacks sufficient details regarding these 30 anticholinergic agents and their respective scores. To enhance the clarity and rigor of the study, it is recommended that the authors provide a more comprehensive description of the methods employed, including the criteria used for selecting these drugs. Including this information would significantly strengthen the foundation of the results and conclusions presented in the paper.
2) In this study, the harmonized procedures for each drug class made it possible to structure the PI and thus prompt the deprescription or replacement of drugs with a high anticholinergic load (P8, LL222-224). However, the specific intervention methods of the pharmacists are not clearly articulated. For example, it remains ambiguous whether the pharmacists recommend the discontinuation of the targeted anticholinergic drugs or suggest alternative medications with lower anticholinergic burden scores. To strengthen the results and conclusions of this research, it is essential to provide a detailed description of the pharmacists' intervention methods. Clarifying these methods would enhance the overall rigor and applicability of the findings.
3) p-value=0.0115, 95%CI [0.2173; 1.6715] (P5, LL179) This confidence interval [0.2173; 1.6715] includes 1, so it is interpreted as meaning that there is no statistically significant difference. Please check.
Author Response
Comment 1: In this study, a pharmacist analyzed the alerts generated by the CDSS for 30 targeted anticholinergic drugs (P1, LL16-17). However, the Methods section lacks sufficient details regarding these 30 anticholinergic agents and their respective scores. To enhance the clarity and rigor of the study, it is recommended that the authors provide a more comprehensive description of the methods employed, including the criteria used for selecting these drugs. Including this information would significantly strengthen the foundation of the results and conclusions presented in the paper.
Response 1: Thank you for pointing this out. To clarify how we selected the drugs, we now write (line 106):
“We selected drugs with different scores (1, 2 and 3) on the basis of drug classes allowing therapeutic reassessment without necessarily requiring specialist advice (for example antihistamines) or for which therapeutic reassessment is carried out in routine practice (for example psychotropic drugs). On the other hand, drugs that are not included are in thera-peutic classes that require specialist advice, such as drugs that require a titration phase before therapeutic balance can be achieved (for example immunosuppressants or anti-epileptics)”.
Comment 2: In this study, the harmonized procedures for each drug class made it possible to structure the PI and thus prompt the deprescription or replacement of drugs with a high anticholinergic load (P8, LL222-224). However, the specific intervention methods of the pharmacists are not clearly articulated. For example, it remains ambiguous whether the pharmacists recommend the discontinuation of the targeted anticholinergic drugs or suggest alternative medications with lower anticholinergic burden scores. To strengthen the results and conclusions of this research, it is essential to provide a detailed description of the pharmacists' intervention methods. Clarifying these methods would enhance the overall rigor and applicability of the findings.
Response 2: We agree with you. The course of action recommended to the physician is given in the text (line 113). Deprescribing was proposed for each PI. For certain drug classes, an alternative was suggested when the indication required drug treatment.
Comment 3: p-value=0.0115, 95%CI [0.2173; 1.6715] (P5, LL179) This confidence interval [0.2173; 1.6715] includes 1, so it is interpreted as meaning that there is no statistically significant difference. Please check.
Response 3: We agree with you and apologize for that mistake. We asked a statistician to check and correct the analysis by applying a Kruskal Wallis test to the study’s primary endpoint. A confidence interval is not necessary for this test. This is now explained in the Methods (line 157), and we have corrected the results (line 205).
Reviewer 3 Report
Comments and Suggestions for Authors
The authors investigated the impact of computerized clinical decision support system (CDSS) with an alert management procedure on reducing anticholinergic load in geriatric patients. This study is noteworthy as majority of geriatric pateints are at elevated riks of receiving anticholinergic prescriptions. However, I recommend some modifications.
1. Please add more information related to CDSS. What factors does CDSS detect? describe some working processes.
2. It would be nice to provide a figure on the study procedures regarding CDSS intervention and control.
3. Line 89+. What are the standard for potentially reducible?
4. Please provide statistical results for Table 1. ARe there any difference between interventional and control group?
5. Please add more details on 17 PI not accepted. What are the reasons for the rejection?
6. Anticholinergic adverse events rates seems to be lower in control group. Please elaborate the potential reasons for this results in the discussion
Author Response
Comment 1: Please add more information related to CDSS. What factors does CDSS detect? describe some working processes.
Response 1: Thank you for this comment. We now describe this in the text (line 64).
Comment 2: It would be nice to provide a figure on the study procedures regarding CDSS intervention and control.
Response 2: We agree that an illustrative explanation would be useful and so have added Figure 1 (describing the working processes) to the text (line 132).
Comment 3: Line 89+. What are the standard for potentially reducible?
Response 3: We have clarified this aspect (line 98) but the criteria are detailed further on in the text (line 106), following Reviewer 2’s request.
Comment 4: Please provide statistical results for Table 1. Are there any difference between interventional and control group?
Response 4: Thank you for pointing this out. The corresponding p values have been added to Table I.
Comment 5: Please add more details on 17 PI not accepted. What are the reasons for the rejection?
Response 5: We agree that it would be useful to know why PIs were rejected. Unfortunately, prescribers did not have to explain why they refused a PI. A new reference has nevertheless been cited (line 225).
Comment 6: Anticholinergic adverse events rates seems to be lower in control group. Please elaborate the potential reasons for this results in the discussion
Response 6: The two cohorts were separate in time and were not deliberately composed of patients with different characteristics. The two cohorts did not differ significantly in terms of the total number of anticholinergic drugs and the number of anticholinergic drugs with a high score drugs. We cannot see an obvious reason for the intergroup difference in the adverse event rate.
Round 2
Reviewer 1 Report
Comments and Suggestions for Authors
The manuscript has been improved by authors. Nevertheless, in my opinion, they should fullfil Comment 14 in a better way, explaining briefly anticholinergic drugs' clinical indications (eventually with a Table summarizing all drugs with anticholinergic effect) and discussing the role of pharmacokinetic interactions. Using one scale only is a relevant limitation, but the authors clearly state it at the end of the manuscript and readers are aware of it.
Author Response
Comment 1: The manuscript has been improved by authors. Nevertheless, in my opinion, they should fullfil Comment 14 in a better way, explaining briefly anticholinergic drugs' clinical indications (eventually with a Table summarizing all drugs with anticholinergic effect)
Response 1: We have described conditions on which anticholinergic drugs are prescribed : "management of various conditions, including depression, epilepsy seizures, neuropathic pain, asthma, overactive bladder and ophthalmic disorders" (line 39).
Comment 2: and discussing the role of pharmacokinetic interactions
Response 2: To adress comment 14, we have added 2 additional references (21 and 22, line 266) on pharmacokinetic interactions. Other references were not clinically relevant to cite in the manuscript. Otherwise, the main objective of this manuscript was to evaluate the impact of the Clinical Decision Support System in helping to formulate pharmaceutical interventions. It is an innovative tool in practice. PK/PD interactions for anticholinergics are not focus of this work. We hope that the edits made in the discussion will be sufficient for publication.
Reviewer 2 Report
Comments and Suggestions for Authors
I see no points that need to be revised, and I judge that the current content is sufficient to be published.
Therefore, I strongly recommend that this paper be accepted as is.
Author Response
We are delighted that revisions convinced the reviewer